# The Prognostic Value of Tumor Fibrosis in Patients Undergoing Hepatic Metastasectomy for Colorectal Cancer: A Retrospective Pooled Analysis

**DOI:** 10.3390/cancers17111870

**Published:** 2025-06-03

**Authors:** Xavier Hernández-Yagüe, Santiago López-Ben, Joan Martínez-Sancho, Maria Rosa Ortíz-Durán, Margarida Casellas-Robert, Ana Aula-Olivar, Cristina Meléndez-Muñoz, Maria Buxó Pujolràs, Bernardo Queralt-Merino, Joan Figueras i Felip

**Affiliations:** 1Department of Medical Oncology, Catalan Institute of Oncology, Doctor Josep Trueta University Hospital, Av. França s/n, 17007 Girona, Spain; bqueralt@iconcologia.net; 2Hepatobiliary and Pancreatic Surgery Unit, Department of Surgery, Dr. Josep Trueta Hospital, IdIBGi, 17007 Girona, Spain; slopezben.girona.ics@gencat.cat (S.L.-B.); mcasellas.girona.ics@gencat.cat (M.C.-R.); 3Unitat de Estadística, Institut de Recerca Biomèdica de Girona (IDIBGi), 17007 Girona, Spain; estadistica@idibgi.org (J.M.-S.); mbuxo@idibgi.org (M.B.P.); 4Pathology Department, Catalan Institute of Health, Hospital Josep Trueta, School of Medicine, University of Girona (UdG), 17007 Girona, Spain; rortizduran.girona.ics@gencat.cat (M.R.O.-D.); aaula.girona.ics@gencat.cat (A.A.-O.); cmelendezm.girona.ics@gencat.cat (C.M.-M.); 5Researcher Affiliated with the Doctoral Program at the University of Girona, 17007 Giona, Spain; jfigueras.net@icloud.com

**Keywords:** liver metastases, colorectal cancer, pathological response, fibrosis

## Abstract

Surgical resection of liver metastases (M1) has shown a clear survival benefit in patients with metastatic colorectal cancer (mCRC). Several prognostic factors have been linked to differences in survival among these patients, including pathological tumor response. Tumor fibrosis in resected liver metastases has been associated with improved outcomes across different classifications, though no consistent cut-off for prognostic value has been established. This study aims to evaluate the likelihood of achieving ≥40% fibrosis following preoperative chemotherapy combined with either anti-epidermal growth factor receptor (anti-EGFR) or anti-vascular endothelial growth factor (anti-VEGF) therapy and its association with overall survival. Additionally, the study compares the prognostic value of two pathological response classifications: Poultsides vs. Rubbia–Brandt. Final outcomes aim to determine whether chemotherapy plus anti-EGFR leads to greater fibrosis and better prognosis than anti-VEGF. The study also seeks to validate the ≥40% fibrosis threshold as a more practical prognostic tool than the two-category Rubbia–Brandt classification.

## 1. Introduction

Colorectal cancer (CRC), including anal cancer, represents a significant global health concern. In 2020, an estimated 1.9 million new cases and 935,000 deaths were reported, accounting for approximately 10% of all cancer cases and fatalities worldwide. In Spain, the observed survival rate between 2008 and 2013 was approximately 52.1% [1]. The prognosis of CRC is strongly influenced by the presence of metastases (M1), most commonly in the liver, which plays a critical role in determining patient outcomes following surgical treatment for metastatic disease.

The surgical resection of hepatic metastases from metastatic colorectal cancer (mCRC), whether open or laparoscopic, has been consistently associated with an overall survival benefit over time [2,3,4,5,6].

Neoadjuvant chemotherapy (NAC) has demonstrated beneficial effects in patients with unresectable or borderline resectable mCRC [7,8,9]. Studies have highlighted the prognostic significance of pathological responses to NAC [10,11,12,13], with the Rubbia–Brandt classification [10] being the most widely used system for categorizing these responses into five tumor regression grades (TRG). Notably, multivariate analyses suggest that TRG is an independent prognostic factor for both disease-free survival (DFS) and overall survival (OS) [10]. This finding underscores the significance of TRG in predicting patient outcomes.

Although no universally accepted threshold for fibrosis or residual tumor exists to differentiate tumor regression grade (TRG) categories, a study conducted by Poultsides et al. has identified a fibrosis level of ≥40% as an independent prognostic marker in the context of NAC for liver metastases from mCRC [14]. However, the reliability of this threshold remains contentious, as it was established from analyses employing various fibrosis cut-offs. This raises concerns regarding potential statistical bias in designating ≥40% as a significant predictor. This classification has come to be known as the Poultsides classification.

This study included only patients who underwent NAC—either chemotherapy alone or in combination with bevacizumab—prior to liver metastasis surgery. Interestingly, only 18.8% of patients treated with bevacizumab exhibited fibrosis ≥40%. In contrast, a subsequent study comparing fibrosis levels across different treatment groups—chemotherapy alone, chemotherapy with bevacizumab, and chemotherapy with cetuximab—revealed a significantly higher incidence of fibrosis ≥40% in the cetuximab-treated group (45.71%) compared to the bevacizumab group (24.4%) [15].

Given the potential prognostic value of increased fibrosis and its higher prevalence in cetuximab-treated patients, further evaluation of cetuximab in combination with chemotherapy as a neoadjuvant treatment strategy is warranted. The simplicity of fibrosis assessment methods—such as reticulin staining, trichrome staining, and hematoxylin-eosin staining—could facilitate its integration as a readily available prognostic marker in clinical practice. Further studies are necessary to validate these findings and optimize treatment strategies to improve survival and disease progression outcomes in mCRC patients.

## 2. Materials and Methods

### 2.1. Hypothesis

The primary objective of this study is to demonstrate that NAC combined with cetuximab or panitumumab (anti-epidermal growth factor receptor, or anti-EGFR) induces a higher degree of fibrosis (≥40%) compared to NAC with bevacizumab (anti-vascular endothelial growth factor, or anti-VEGF). Moreover, the primary objective of this study is to investigate differences in overall survival based on the presence or absence of ≥40% fibrosis in liver metastases resected after NAC, as proposed by the Poultsides classification [14]. The secondary objective is to compare the prognostic significance of the Poultsides classification, which considers fibrosis ≥ 40% as a favorable prognostic indicator, with that of the Rubbia–Brandt classification, which has been reclassified into two categories based on TRG.

### 2.2. Study Population

The current study retrospectively included mCRC patients who underwent liver resection between September 2005 and January 2023 at the Hepatopancreatobilary Surgery Department of University Hospital of Girona, Dr. Josep Trueta. All data were collected and analyzed retrospectively. The study was approved by the Ethics Committee of Hospital Dr. Josep Trueta on 23 July 2019 and was registered under the code 2019.130. It was conducted in accordance with the ethical principles outlined in the Declaration of Helsinki. The inclusion criteria were as follows: (1) pathologically confirmed CRC; (2) considered a resectable or potentially resectable liver metastases by a multidisciplinary team (MDT) at diagnosis; (3) Having received at least four cycles of neoadjuvant chemotherapy and anti-EGFR or anti-VEGF targeted therapy. (4) Having undergone radical surgery for hepatic metastases (liver surgery = cR0), regardless of whether surgery for extrahepatic metastases was performed. According to the criteria, this retrospective observational study included 108 patients (Figure 1).

Hepatopathy prior to hepatic surgery was classified according to the following criteria: 0 (none) = no history of liver disease; 1 (mild) = history of abnormal liver enzyme levels with negative hepatitis virus serology and no signs of chronic liver disease; 2 (moderate) = history of abnormal liver enzyme levels with positive or negative hepatitis virus serology, with clinical or imaging signs of chronic liver disease but without cirrhosis; 3 (severe) = liver cirrhosis of any etiology, classified as Child–Pugh A.

The clinical risk score (CRS) is based on the classification by Fong et al. [16], which includes five categories. For the purposes of this study, these were reclassified into two groups: CRS 0–2 (low risk) and CRS 3–5 (intermediate or high risk).

The lymph node ratio (LNR) is defined as the ratio of positive locoregional lymph nodes to the total number of lymph nodes retrieved in the colectomy or rectal resection specimen.

The pT or pN stage refers to patients who underwent primary tumor resection prior to receiving neoadjuvant chemotherapy (NAC), such as those with metachronous liver metastases. In contrast, the ypT or ypN stage refers to patients who had primary tumor resection after receiving NAC, typically in the context of metachronous metastatic colorectal cancer (mCRC).

### 2.3. Statistics

Sample Size Calculation: Based on the data from Stremitzer’s study [15], in which mCRC patients treated with cetuximab achieved fibrosis ≥40% in 45.7% of cases (16/35), while those treated with bevacizumab achieved fibrosis ≥40% in 24.2% of cases (24/99), a sample size calculation was performed. Assuming an alpha error of 0.05 and a statistical power of 0.80, a total of 52 patients in the anti-EGFR group (Cohort 1 or EGFR inhibitor) and 52 in the anti-VEGF (Cohort 2 or VEGF inhibitor) group is required to detect a significant difference in the fibrosis rate (45% vs. 25%) [17,18,19,20,21].

Statistical Analysis: Categorical variables will be described using absolute and relative frequencies (%) and analyzed using the Chi-square test or Fisher’s exact test, as appropriate. Quantitative variables will be summarized using the mean and standard deviation (SD) or the median and interquartile range [IQR], depending on data distribution. Normality will be assessed using the Shapiro–Wilk test. Quantitative variables with a normal distribution will be analyzed using the Student’s *t*-test for independent samples and one-way analysis of variance (ANOVA), while non-normally distributed variables will be analyzed using the Mann–Whitney U test or the Kruskal–Wallis test. Logistic regression modeling will be used to assess the factors associated with fibrosis ≥40%. First, a univariate analysis will be performed, and significant and clinically relevant factors will be selected for inclusion in the multivariate regression model. For survival analysis, Kaplan–Meier curves will be generated and compared using the log-rank test. If necessary, Cox regression analysis will be performed to evaluate the effect of independent and/or confounding variables. Hazard ratios (HR) and their corresponding 95% confidence intervals (CI) will be reported. To verify compliance with the proportional hazards assumption in the survival analysis using the Cox regression model, a Schoenfeld residuals analysis was performed on the univariate model. Only variables that met the proportional hazards assumption (*p* > 0.05 in the Schoenfeld test) were included in the multivariate Cox regression analysis. For the comparative analysis of the prognostic informative value of survival models, the Akaike information criterion (AIC) has been employed.

All analyses will be conducted using SPSS and R statistical software, with a significance level set at 5%.

### 2.4. Pathological Specimen Analysis

The pathological examination of liver resection specimens was performed in accordance with guidelines established by the Spanish group [22]. All nodules ≤ 2 cm in diameter were fully embedded. For nodules measuring between 2 and 5 cm, at least one complete panoramic section was required, accompanied by an inclusion diagram. For nodules larger than 5 cm, an additional tissue block was included for each additional centimeter beyond this threshold. To adequately capture tumor heterogeneity, both central and peripheral regions of large nodules were sampled. Additionally, non-tumor liver tissue was obtained at least 2 cm away from the metastasis whenever possible. If this distance was not feasible, the most distant available sample was collected.

Histological staining protocols included standard hematoxylin-eosin staining. Non-tumor liver tissue was stained with at least Masson’s trichrome and reticulin stains, and additional staining techniques such as periodic acid–Schiff (PAS) and Perl’s iron stain were utilized when appropriate.

The macroscopic evaluation of resected specimens involved quantifying the number of metastases, measuring their maximum diameters, and assessing resection margins. Pathological reports were generated by three expert pathologists (M.R. Ortiz, C. Meléndez, and A. Aula) without a secondary blind review.

The percentage of tumor, fibrosis, and residual necrosis within each metastasis was documented. Based on these parameters, two tumor regression grade (TRG) classifications were applied:

The Rubbia–Brandt classification (recoded into two categories): good tumor response (TRG1, TRG2, and TRG3) vs. poor tumor response (TRG4 and TRG5) and the Poultsides classification, categorized according to fibrosis level: fibrosis ≥40% vs. fibrosis < 40%.

For patients who underwent resection of multiple metastases, the median values of residual tumor, fibrosis, and necrosis across all metastases were used to determine TRG classification.

## 3. Results

### 3.1. Descriptives

Between September 2005 and January 2023, 437 hepatic surgeries were performed, with 378 patients undergoing liver resection for colorectal cancer metastases. Of these, 108 met the study criteria and were evenly divided into two cohorts: 54 patients (50%) received chemotherapy combined with anti-EGFR therapy (Cohort 1 or EGFR inhibitor), and 54 patients (50%) received chemotherapy combined with anti-VEGF therapy (Cohort 2 or VEGF inhibitor) (Figure 1).

#### 3.1.1. Baseline Clinical and Tumor Characteristics

Baseline clinical and tumor-related preoperative characteristics for both cohorts are summarized in Table 1. The groups were generally well balanced, except for a significantly higher frequency of RAS or BRAF mutations in Cohort 2 (74%) compared to Cohort 1 (7.41%) (*p* < 0.001) and a higher Ca 19.9 level at metastasis diagnosis in Cohort 2 (median: 115 U/mL [IQR: 22.4–650] vs. 34.4 U/mL [IQR: 9.6–149]; *p* = 0.032).

The median age at liver surgery was 63 years [IQR: 56.0–70.2]. Most patients had left-sided primary tumors (84.3% vs. 15.7%) and had undergone primary tumor resection (94.4%). The majority of resected primary tumors were locally advanced (pT3–pT4: N = 53 (93%) or ypT3–ypT4: N = 39 (90.7%)) and frequently had nodal involvement (pN1–pN2: N = 45 (79%) or ypN1–ypN2: N = 27 (62.8%)).

Synchronous metastases were present in 84 patients (77.8%), while 28 patients (25.9%) had extrahepatic metastases, most commonly in the lungs (N = 13, 46.4%).

At diagnosis, the median number of liver metastases was 4 [IQR: 2–7], and the median number of affected hepatic segments was 4 [IQR: 2–5]. Large liver metastases (≥5 cm) were observed in 47 patients (46.1%). Initially, 70.4% (N = 76) of patients were deemed technically unresectable by a multidisciplinary team, and 61.7% (N = 58) had an intermediate or high Clinical Risk Score (CRS 3–5). A total of 66 patients (61.1%) underwent hepatic metastasis surgery during the 2005 period, compared to 42 patients (28.9%) during the 2014–2023 period.

#### 3.1.2. Preoperative Treatment Characteristics and Radiologic Tumor Response

Preoperative treatment characteristics and radiologic tumor response are summarized in Table 2.

Most patients (N = 69, 63.9%) received oxaliplatin-based chemotherapy, with a median of seven neoadjuvant chemotherapy cycles [IQR: 5–11] and six monoclonal antibody cycles [IQR: 4–9]. The median preoperative chemotherapy dose intensity was 95% [IQR: 90–100].

In terms of radiologic response (RECIST response criteria), the majority of patients (N = 74, 69.8%) achieved either a partial or complete tumor response.

Significant differences were observed in monoclonal antibody dose intensity, which was slightly lower in Cohort 1 (97.13%) compared to Cohort 2 (99.91%); *p* = 0.013, and in preoperative Ca 19.9 levels, which were significantly lower in Cohort 1 (16.5 U/mL) than in Cohort 2 (32.2 U/mL); *p* = 0.003.

#### 3.1.3. Hepatic Surgery Characteristics and Complications

Appendix A summarizes the characteristics of hepatic surgery and postoperative complications. Ninety-day postoperative mortality was significantly lower in the VEGF inhibitor group (N = 0, 0%) compared to the EGFR inhibitor group (N = 9, 16.7%); *p* = 0.003.

No significant differences were observed in surgical approach (types of liver surgery), operative time (liver surgery duration), preoperative portal embolization or ligation, or vascular clamping duration. However, the interval between the last monoclonal antibody dose and surgery was longer in Cohort 2 (6 weeks [IQR: 4–9.75]) than in Cohort 1 (4 weeks [IQR: 3–7]), likely reflecting the well-documented challenges of VEGF inhibitor-associated wound healing and bleeding complications [23].

A significant difference was observed in the incidence of postoperative hemoperitoneum, which was more frequent in Cohort 1 (N = 10, 18.5%) than in Cohort 2 (N = 2, 3.7%); *p* = 0.032. However, no significant differences were found in other postoperative complications, including biliary fistula, intra-abdominal abscess, wound infection, pneumonia, or liver failure.

#### 3.1.4. Postoperative Treatment and Recurrence Characteristics

Regarding postoperative treatment and recurrence Appendix A, significant differences were observed in the administration of adjuvant therapy following hepatic surgery or first-line therapy for patients with persistent, unresected metastases. A greater proportion of patients in Cohort 1 (EGFR inhibitor) did not receive postoperative treatment (N = 11, 23.9%) compared to Cohort 2 (VEGF inhibitor, N = 1, 2.13%); *p* < 0.001, possibly due to the higher postoperative mortality in Cohort 1. However, no significant differences were observed in post-treatment progression-free survival (PFS) or in the incidence of hepatic or pulmonary recurrences.

#### 3.1.5. Pathologic Tumor Response

The median residual fibrosis in resected metastases was significantly higher in the EGFR inhibitor—Cohort 1 (40.0% [IQR: 25.4–53.2]) compared to the VEGF inhibitor—Cohort 2 (20.6% [IQR: 8.07–36.9]); *p* < 0.001. Conversely, the median residual necrosis was lower in Cohort 1 (16.1% [IQR: 5.00–25.0]) than in Cohort 2 (33.0% [IQR: 15.0–50.0]); *p* = 0.001 (Table 3).

When fibrosis was classified using the Poultsides criteria (2-category response), a significantly higher proportion of patients in Cohort 1 exhibited fibrosis ≥40% (N = 29, 53.7%) compared to Cohort 2 (N = 13, 24.1%); *p* = 0.003. Similarly, categorized necrosis ≥40% was more frequent in Cohort 2 (N = 24, 44.4%) than in Cohort 1 (N = 7, 13%); *p* = 0.001.

No significant differences were observed between cohorts in pathologic response grading according to the Rubbia–Brandt (2-category) classification (*p* = 0.438). Additionally, no differences were found in the weight of resected liver tissue, pathological size of the largest metastasis, or margin clearance distance.

#### 3.1.6. Non-Tumor Liver Toxicity

In terms of hepatic toxicity associated with preoperative treatment Appendix A, sinusoidal injury was less frequent in Cohort 2 (VEGF inhibitor), where the majority of patients exhibited mild or no sinusoidal damage (N = 48, 88.9%) compared to Cohort 1 (N = 34, 63.0%); *p* = 0.003. No significant differences were found between the two cohorts regarding liver steatosis (*p* = 0.071), non-alcoholic steatohepatitis (*p* = 0.330), or clinical signs of portal hypertension post-hepatic surgery (*p* = 0.158).

### 3.2. Monoclonal Antibody Use and Fibrosis Degree: Analytics

To further explore factors associated with fibrosis ≥40%, a logistic regression analysis was performed (Table 4), as univariate analysis does not account for interactions among multiple variables influencing tumor fibrosis.

#### 3.2.1. Monoclonal Antibody Use and Fibrosis Degree: Univariate Logistic Regression Analysis

In univariate analysis, the following variables were identified as potential predictors of fibrosis ≥40%: clinical risk score (CRS) (*p* = 0.025), preoperative monoclonal antibody type (*p* = 0.002), primary tumor location (*p* = 0.062) and RAS/BRAF mutations (*p* = 0.006).

#### 3.2.2. Monoclonal Antibody Use and Fibrosis Degree: Multivariate Logistic Regression Analysis

Given the strong collinearity between RAS/BRAF mutations and monoclonal antibody type (as nearly all patients in Cohort 1 (92.6%) were RAS/BRAF wild-type), RAS/BRAF mutations were excluded from the final multivariate model. Monoclonal antibody type was the strongest independent predictor of fibrosis ≥40% (odds ratio (OR) = 4.967 for EGFR inhibitors, 95% CI: 1.885–13.087; *p* = 0.001). CRS was also a significant predictor, with higher CRS scores (3–5) being associated with a lower likelihood of fibrosis ≥40% (OR = 0.377, 95% CI: 0.146–0.975; *p* = 0.044). Primary tumor location did not reach statistical significance in the multivariate model (OR = 4.607 for left-sided tumors, 95% CI: 0.854–24.855; *p* = 0.076), but it was retained due to potential clinical relevance, small sample size, and a *p*-value approaching significance. This multivariate model has a sensitivity of 33.3% and specificity of 89.7%.

#### 3.2.3. Monoclonal Antibody Use and Fibrosis Degree: Multivariate Model Equation

The probability (P) of fibrosis ≥40% can be estimated using the following formula: log (P/1 − P) = −2.174 + (1.603 × VEGF inhibitor = 0 or EGFR inhibitor = 1) + (−0.974 × CRS 0–2 = 0 or CRS 3–5 = 1) + (1.528 × right colon = 0 or left colon = 1) or equivalently P = 1/1 + e(−1 × ((−2.174 + (1.603 × VEGF inhibitor = 0 or EGFR inhibitor = 1) + (−0.974 × CRS 0–2 = 0 or CRS 3–5 = 1) + (1.528 × right colon = 0 or left colon = 1)).

This model defines eight probability groups for fibrosis ≥40% based on tumor location, monoclonal antibody type, and CRS (Table 5).

### 3.3. Overall Survival Analysis

To assess overall survival (OS), we excluded nine patients who experienced early (90-day) postoperative mortality to minimize confounding effects and better isolate the impact of neoadjuvant treatment, surgical interventions, and pathological outcomes on long-term survival.

The median follow-up period was 41.6 months [IQR 0–215], with 34.9 months [IQR 0–171] in Cohort 1 and 48.6 months [IQR 4–215] in Cohort 2. By the end of the study, 88 patients (81.5%) had died, and 82 patients (75.9%) had experienced disease recurrence or progression. Among the deceased, 24 patients (22.2%) died from non-cancer-related causes, including postoperative mortality.

Two patients were lost to follow-up due to relocation, and their mortality and disease progression status remained unknown. For survival analysis, their OS events were censored at the last available follow-up, meaning they contributed prognostic information only up to that point.

Ultimately, 99 patients were included in the overall survival analysis.

Given that a small proportion of patients ultimately did not undergo resection of either extrahepatic metastases or the primary tumor (CR2 = 13; 12%), survival differences were analyzed according to surgical outcome—complete resection (CR0) vs. incomplete resection (CR2). Additionally, survival analysis was performed based on tumor fibrosis groups (<40% vs. ≥40%) and monoclonal antibody associated with chemotherapy (anti-EGFR vs. anti-VEGF) stratified by type of surgery (CR0 or CR2), considering the potential negative impact of incomplete resection (CR2) on patient survival.

Overall survival in the entire cohort showed a trend favoring patients with tumor fibrosis > 40% (median survival = 51 months, 95% CI: 20.90–81.09) compared to those with fibrosis < 40% (median survival = 30 months, 95% CI: 22.83–37.16); log-rank test *p* = 0.036. This difference remained statistically significant when analyzing only patients who underwent CR0 resection of the primary tumor and all metastases (N = 87); log-rank test *p* = 0.028.

When assessing the effect of anti-EGFR vs. anti-VEGF therapy on overall survival in the full cohort, no significant differences were observed (log-rank test *p* = 0.953), with a median survival of 28 months (95% CI: 20.54–35.45) in cohort 1 (anti-EGFR) and 34 months (95% CI: 21.75–46.24) in cohort 2 (anti-VEGF). Similarly, no significant difference was found when the analysis was restricted to patients with CR0 resection; log-rank test *p* = 0.912. Figure 2.

#### 3.3.1. Univariate OS Cox Regression Analysis

Univariate Cox regression Appendix A identified the following variables as potentially associated with improved prognosis: female sex (hazard ratio (HR) = 0.60, 95% CI 0.37–0.97, *p* = 0.033), primary tumor resection (HR = 0.22, 95% CI 0.09–0.56, *p* = 0.009), and fibrosis ≥40% ((Poultsides classification) (HR = 0.61, 95% CI 0.38–0.97, *p* = 0.034)).

Conversely, variables associated with a potentially worse prognosis included the following: lymph node ratio ((LNR) (HR = 3.99, 95% CI 1.25–12.7, *p* = 0.032)), preoperative CEA level (HR = 1.001, 95% CI 1.0004–1.0024, *p* = 0.006), liver metastasis size ≥10 cm (M1 ≥ 10 cm) (HR = 2.34, 95% CI 1.11–4.92, *p* = 0.044), degree of hepatopathy at diagnosis (moderate/severe) (HR = 2.63, 95% CI 1.63–4.26, *p* < 0.001), and resected primary tumor and all metastases (CR2) (HR = 1.96, 95% CI 1.03–3.74, *p* = 0.04).

Although tumor pathological response based on the Rubbia–Brandt classification (2 categories) was not statistically significant in the univariate model, we developed two multivariate overall survival models, incorporating significant univariate predictors and each pathological response classification separately.

#### 3.3.2. Multivariate OS Models

Poultsides Classification Model (Table 6): In the survival model based on the Poultsides classification, the following variables had a positive impact on OS: fibrosis ≥40% (HR = 0.488, 95% CI 0.270–0.880, *p* = 0.014) and female sex (HR = 0.505, 95% CI 0.281–0.909, *p* = 0.019). Conversely, the following variables were associated with worse overall survival: moderate or severe hepatopathy (HR = 2.678, 95% CI 1.535–4.672, *p* < 0.001) and lymph node ratio (LNR) (HR = 4.190, 95% CI 1.222–14.36, *p* = 0.033).

Rubbia–Brandt Classification Model (Table 6): In the multivariate survival model using the Rubbia–Brandt classification, the following variables were associated with better overall survival: TRG responders (TRG1–3) (HR = 0.549, 95% CI 0.321–0.938, *p* = 0.028) and female sex (HR = 0.531, 95% CI 0.294–0.960, *p* = 0.031). In contrast, the following variables were associated with worse overall survival: moderate or severe hepatopathy (HR = 2.821, 95% CI 1.612–4.936, *p* < 0.001) and preoperative CEA level (HR = 1.002, 95% CI 1.000–1.003, *p* = 0.029).

Complete resection of the primary tumor and all metastases (CR0 vs. CR2) did not demonstrate predictive values for survival in the multivariate analysis in either the Poultsides or Rubbia–Brandt pathological response models (*p* = 0.901 and *p* = 0.922, respectively).

Both models were compared using the Akaike information criterion (AIC) to assess their relative explanatory power. The AIC for the Poultsides model is 407.284 and the AIC value for the Rubbia–Brandt model is 408.493. The small difference in AIC values suggests that both models provide equivalent predictive value, with no clear preference for one over the other.

A calculator has been developed (in Excel format) to estimate survival probability (%) based on both pathological tumor regression models (Poultsides and Rubbia–Brandt), incorporating the significant variables identified in each model as well as their corresponding hazard ratios for each estimated time point Appendix A.

## 4. Discussion

Several classification systems have been proposed for assessing pathological response in resected hepatic metastases from colorectal cancer (CRC). Many of these classifications are subject to interobserver variability among pathologists or involve overlapping categories that are difficult to distinguish. This study aims to determine whether the pathological response in resected metastases differs between anti-VEGF and anti-EGFR therapies, focusing on fibrosis ≥40% (Poultsides classification) and the Rubbia–Brandt criteria (in two categories). Additionally, we assess the prognostic value of each classification within a multivariate overall survival model.

Patients were included from the 2005–2023 period to fulfill the sample size requirements defined in the initial study design. We recognize that this extended time frame encompasses significant advancements in both surgical and systemic treatments for metastatic colorectal cancer (mCRC). As such, this temporal variability may constitute a limitation of the study, particularly when interpreting overall survival outcomes across the cohort. The characteristics of our cohort highlight a population with poor prognosis: 70.4% of patients were initially deemed unresectable by a multidisciplinary team, 61.7% had high or intermediate clinical risk scores (CRS 3–5), and 46.1% had metastases larger than 5 cm. Notably, in the anti-EGFR group (Cohort 1), early postoperative mortality was 16.1%, compared to 0% in the anti-VEGF group (Cohort 2). This was primarily due to hemoperitoneum (44.4% vs. 8.08%) and ascites (66.7% vs. 9.09%), leading to hepatic failure (66.7% vs. 9.09%). This discrepancy may be explained by the protective effects of bevacizumab (anti-VEGF) on hepatic sinusoids, reducing portal vascular resistance in patients treated with oxaliplatin and lowering the risks of intraoperative bleeding and portal hypertension-related ascites, as previously reported [24,25,26,27]. Significant differences were observed in the proportion of RAS/BRAF wild-type patients between cohort 1 and cohort 2 (92.6% vs. 26%, respectively; *p* < 0.001), reflecting the exclusive indication of anti-EGFR therapy, as per current clinical guidelines, for patients with RAS/BRAF wild-type tumors. This finding supports the conclusion that any potential benefit of anti-EGFR therapy over anti-VEGF therapy would be restricted, theoretically, to the RAS/BRAF wild-type population. The mutational status itself may influence overall survival and pathological response outcomes, as these represent molecularly distinct populations, introducing a potential molecular selection bias in the study.

Regarding our primary objective, a Chi-square analysis revealed a significantly higher proportion of fibrosis ≥40% (according to the Poultsides classification) in patients treated with chemotherapy combined with anti-EGFR therapy (in a RAS/BRAF wild-type enriched population; 92.6%) compared to those receiving chemotherapy with anti-VEGF therapy (in a RAS/BRAF mutant enriched population; 26%); 53.7% vs. 24.1%, respectively; *p* = 0.003. A dedicated analysis of RAS/BRAF wild-type patients would be necessary to determine the specific benefit, in terms of fibrosis ≥40%, and to support a clear therapeutic recommendation favoring either anti-EGFR or anti-VEGF treatment. However, no significant differences were found when pathological response was assessed using the Rubbia–Brandt criteria (*p* = 0.438). The results of the logistic regression analysis (both univariate and multivariate, Table 4) revealed several significant associations with the likelihood of having fibrosis ≥40%. Specifically, anti-EGFR treatment (in a RAS/BRAF wild-type enriched population) was positively associated with an increased likelihood of fibrosis ≥40%, (OR = 4.967, 95% CI: 1.885–13.087; *p* = 0.001). Additionally, the presence of a left-sided primary tumor was also associated with a higher likelihood of fibrosis ≥40%, though this association was not statistically significant, as evidenced by an OR of 4.607 (95% CI: 0.854–24.855; *p* = NS). On the other hand, a clinical risk score (CRS) of 3–5 was inversely associated with the likelihood of fibrosis ≥40%, suggesting a resistance effect to the development of fibrosis in high-risk patients (OR = 0.377, 95% CI: 0.146–0.975; *p* = 0.044).

These findings may be influenced by selection bias, as most patients receiving anti-EGFR therapy were RAS/BRAF wild-type (92.6%), compared to only 26% in the anti-VEGF group (Table 1). Consequently, these results do not support a universal preference for anti-EGFR over anti-VEGF therapy in RAS/BRAF wild-type patients based solely on the probability of fibrosis ≥40%. Overall, right-sided CRC liver metastases are less likely to exhibit fibrosis ≥40% compared to left-sided CRC metastases. However, in resected liver metastases from right-sided CRC, anti-EGFR therapy increases the likelihood of fibrosis ≥40% (17.5–36.1%) compared to anti-VEGF therapy (4.1–10.2%) (Table 5), according to a multivariate logistic regression model. This finding aligns with the guidelines from the Pan-Asian and European Society for Medical Oncology (ESMO) [28,29], which recommend considering anti-EGFR therapy for potentially resectable right-sided colorectal cancer metastases in patients with wild-type RAS/BRAF, although our recommendation is based on pathological response.

In our view, strategies aimed at enhancing the pathological response, particularly fibrosis, should be explored to improve overall survival in patients undergoing resection of colorectal cancer liver metastases. For this analysis, patients with early postoperative mortality (N = 9) were excluded, as their outcomes were more reflective of surgical complications or preoperative clinical conditions rather than the impact of treatment strategy on survival.

Univariate overall survival analysis suggested that fibrosis ≥40% (Poultsides classification) could be a significant prognostic factor (fibrosis ≥40% vs. <40%, HR = 0.61, 95% CI: 0.38–0.97; *p* = 0.034). However, overall survival was independent of the monoclonal antibody used (anti-VEGF vs. anti-EGFR; HR = 0.99, 95% CI: 0.63–1.55; *p* = 0.96, Appendix A). This underscores the complexity of metastatic CRC, where overall survival depends on multiple factors beyond the type of biological therapy administered.

In our cohort, univariate survival analysis identified female sex as a favorable prognostic factor for overall survival (HR = 0.6, 95% CI: 0.37–0.97; *p* = 0.033), as well as primary tumor resection (HR = 0.22, 95% CI: 0.09–0.56; *p* = 0.009). However, the survival benefit associated with tumor resection may reflect selection bias, as patients with better prognoses are more likely to undergo surgery. Adverse prognostic factors included elevated Preoperative carcinoembryonic antigen (CEA) levels (HR = 1.001, 95% CI: 1.0004–1.0024; *p* = 0.006), higher lymph node ratio (HR = 3.99, 95% CI: 1.25–12.7; *p* = 0.032), the presence of liver metastases ≥10 cm (HR = 2.34, 95% CI: 1.11–4.92; *p* = 0.044), and moderate-to-severe preoperative hepatopathy (HR = 2.63, 95% CI: 1.63–4.26; *p* < 0.001) Appendix A.

Regarding the secondary objective—assessing the prognostic value of pathological response using the Poultsides classification (fibrosis ≥40% vs. <40%) compared to the Rubbia–Brandt classification (categorized into two groups)—we constructed two multivariate overall survival models. Both models incorporated the same prognostic variables that were significant in the univariate analysis, including gender, lymph node ratio, pre-operative CEA levels, and preoperative hepatopathy, while substituting one pathological response classification for the other. The primary tumor resection variable (yes vs. no) was excluded due to its potential confounding effect.

The analysis demonstrated that both models provided equivalent prognostic information, as indicated by nearly identical Akaike information criterion (AIC) values. In the absence of direct comparative studies between these two classification systems, our findings suggest that assessing pathological response using the Poultsides classification yields prognostic information comparable to that obtained with the Rubbia–Brandt classification in patients undergoing surgical resection of liver metastases (M1) from mCRC.

Given the increasing use of semi-automated histopathological analysis [30], the Poultsides classification, with its specific fibrosis cutoff, may offer a more objective and reproducible approach for assessing pathological response. Its standardized threshold could facilitate its adoption in clinical practice and research, particularly in studies evaluating treatment response in hepatic metastases from mCRC.

These findings highlight the need for further validation through prospective studies, particularly to determine whether fibrosis-based classifications can enhance treatment stratification and guide therapeutic decisions in metastatic colorectal cancer.

One of the main limitations of the present study is its retrospective nature. The data have been collected over an extended period during which both surgical and non-surgical management strategies have evolved. These changes may have influenced overall survival outcomes. Additionally, although pathological assessments were conducted by expert digestive pathologists, no blinded review of the specimens was performed, which may limit the external validity of the findings.

## 5. Conclusions

While anti-EGFR therapy was associated with a higher likelihood of tumor fibrosis ≥40% compared to anti-VEGF therapy, this did not result in improved overall survival among patients undergoing resection of hepatic M1 metastases from metastatic colorectal cancer (mCRC). It is important to consider that the indication for anti-EGFR therapy is restricted to patients without RAS or BRAF mutations. In the present study, there was an imbalance in the distribution of these mutations between the two cohorts, with a higher prevalence of RAS/BRAF mutations in cohort 2 (anti-VEGF group). Therefore, no definitive conclusions can be drawn regarding differences in fibrosis ≥40% between anti-EGFR and anti-VEGF therapy within the RAS/BRAF wild-type population. A dedicated study involving exclusively RAS/BRAF wild-type patients would be necessary to establish whether anti-EGFR therapy is superior to anti-VEGF therapy in terms of inducing tumor fibrosis ≥40%. Fibrosis ≥40% was identified as an independent prognostic factor for survival in patients with resected hepatic metastases, regardless of the type of monoclonal antibody used prior to surgery. Furthermore, fibrosis ≥40% provided prognostic information comparable to that of the Rubbia–Brandt classification in a logistic regression model. Our model is based on data from a single center; therefore, both internal and external validation are necessary to support its future clinical applicability in terms of prognostic or predictive value for survival.

The standardized threshold used in the Poultsides classification, combined with its potential for easier, more objective, and reproducible assessment, suggests that it may offer a practical and reliable alternative for evaluating pathological response in future studies and clinical practice.

All listed authors affirm their substantial contributions to the conception, design, analysis, and/or interpretation of the data presented in this study. Additionally, all authors were involved in drafting and critically revising the manuscript and approved its final version for publication. None of the authors received financial or non-financial compensation for their involvement in this research. All authors declare no conflicts of interest related to their contribution to the study or the preparation of the manuscript.

## Figures and Tables

**Figure 1 cancers-17-01870-f001:**
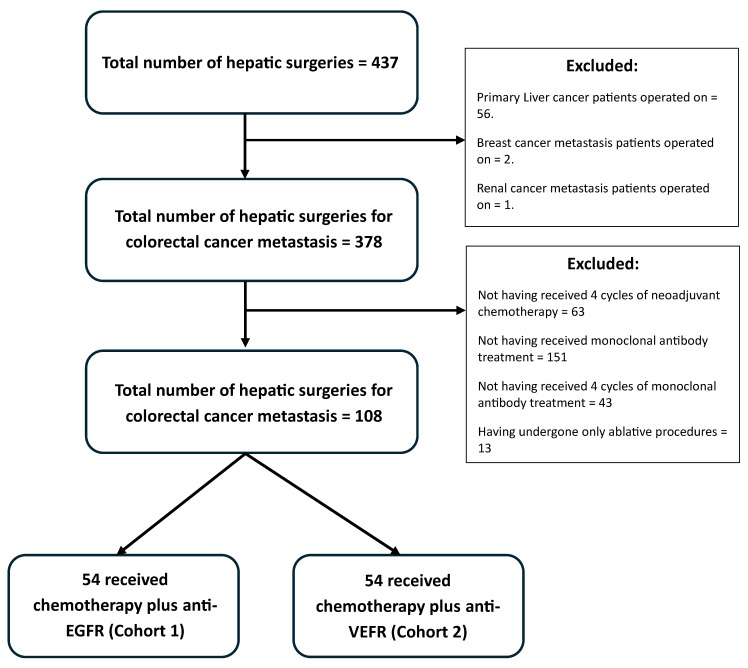
Patient selection flowchart.

**Figure 2 cancers-17-01870-f002:**
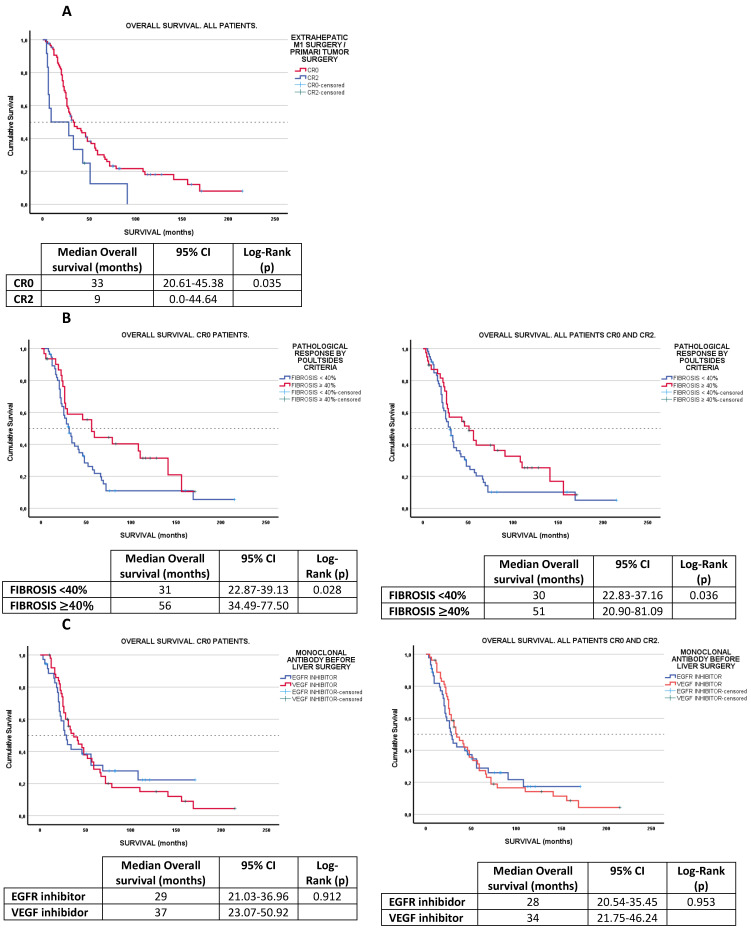
Overall Survival Plots. **Kaplan-Meier survival plots**: (**A**): CR0 vs. CR2 comparison. (**B**): fibrosis < 40% vs. ≥40%. (**C**): EGFR inh vs. VEGF inh comparison.

**Table 1 cancers-17-01870-t001:** Clinical and Colorectal Cancer Characteristics.

	[ALL] N = 108	Cohort 1 (EGFR INHIBITOR) N = 54	Cohort 2 (VEGF INHIBITOR) N = 54	*p*-Value
**GENDER:**				0.692
MALE	67 (62.0%)	35 (64.8%)	32 (59.3%)	
FEMALE	41 (38.0%)	19 (35.2%)	22 (40.7%)	
**AGE AT LIVER SURGERY**	63.0 [56.0;70.2]	63.0 [55.2;71.0]	64.5 [57.0;70.0]	0.629
**PRIMARY TUMOR SIDE:**				0.597
RIGHT COLON	17 (15.7%)	7 (13.0%)	10 (18.5%)	
LEFT COLON	91 (84.3%)	47 (87.0%)	44 (81.5%)	
**OPERATED PRIMARY TUMOR:**				1.000
NO	6 (5.56%)	3 (5.56%)	3 (5.56%)	
YES	102 (94.4%)	51 (94.4%)	51 (94.4%)	
**pTNM (OPERATED PATIENTS BEFORE SYSTEMIC TREATMENT)**				
**pT:**				0.016
pT1	3 (5.26%)	3 (10.3%)	0 (0.00%)	
pT2	1 (1.75%)	0 (0.00%)	1 (3.57%)	
pT3	38 (66.7%)	15 (51.7%)	23 (82.1%)	
pT4	15 (26.3%)	11 (37.9%)	4 (14.3%)	
**pN:**				0.763
pN0	12 (21.1%)	7 (24.1%)	5 (17.9%)	
pN1	23 (40.4%)	12 (41.4%)	11 (39.3%)	
pN2	22 (38.6%)	10 (34.5%)	12 (42.9%)	
**ypTNM (OPERATED PATIENTS AFTER SYSTEMIC TREATMENT)**				
**ypT:**				1.000
ypT0	1 (2.33%)	1 (4.55%)	0 (0.00%)	
ypT1	1 (2.33%)	1 (4.55%)	0 (0.00%)	
ypT2	2 (4.65%)	1 (4.55%)	1 (4.76%)	
ypT3	32 (74.4%)	16 (72.7%)	16 (76.2%)	
ypT4	7 (16.3%)	3 (13.6%)	4 (19.0%)	
**ypN:**				0.190
ypN0	16 (37.2%)	8 (36.4%)	8 (38.1%)	
ypN1	18 (41.9%)	7 (31.8%)	11 (52.4%)	
ypN2	9 (20.9%)	7 (31.8%)	2 (9.52%)	
**LYMPH NODE RATIO (LNR) ***	0.06 [0.00;0.14]	0.07 [0.00;0.17]	0.06 [0.00;0.14]	0.717


	**[ALL] N = 108**	**Cohort 1 (EGFR INHIBITOR) N = 54**	**Cohort 2 (VEGF INHIBITOR) N = 54**	***p*-value**
**RAS_BRAF_MUTATIONS:**				<0.001
NO RAS/BRAF MUTATIONS	63 (60.6%)	50 (92.6%)	13 (26.0%)	
RAS OR BRAF MUTATIONS	41 (39.4%)	4 (7.41%)	37 (74.0%)	
**CEA AT THE METASTATIC DIAGNOSIS (ng/mL)**	43.2 [18.0;114]	38.1 [14.9;92.1]	48.9 [21.6;138]	0.466
**CA 19.9 AT THE METASTATIC DIAGNOSIS (UI/mL)**	56.5 [14.3;346]	34.4 [9.60;149]	115 [22.4;650]	0.032
**SYNCHRONOUS vs. METACHRONOUS LIVER METASTASES:**				
SYNCHRONOUS	84 (77.8%)	41 (75.9%)	43 (79.6%)	
METACHRONOUS	24 (22.2%)	13 (24.1%)	11 (20.4%)	
**EXTRAHEPATIC LIVER METASTASES:**				0.826
NO	80 (74.1%)	39 (72.2%)	41 (75.9%)	
YES	28 (25.9%)	15 (27.8%)	13 (24.1%)	
**NUMBER OF LIVER METASTASES**	4.00 [2.00;7.00]	4.00 [2.00;6.00]	4.00 [3.00;7.75]	0.478
**NUMBER OF LIVER SEGMENTS AFFECTED**	4.00 [2.00;5.00]	3.50 [2.00;4.00]	4.00 [2.00;5.00]	0.263
**SIZE OF LIVER METASTASES ≥ 5 cm:**				0.160
M1 < 5 CM	55 (53.9%)	24 (46.2%)	31 (62.0%)	
M1 ≥ 5 CM	47 (46.1%)	28 (53.8%)	19 (38.0%)	
**SIZE OF LIVER METASTASES ≥ 10 cm:**				1.000
M1 < 10 CM	91 (89.2%)	46 (88.5%)	45 (90.0%)	
M1 ≥ 10 CM	11 (10.8%)	6 (11.5%)	5 (10.0%)	
**INITIAL RESECTION CRITERIA:**				0.833
UNRESECTABLE	76 (70.4%)	37 (68.5%)	39 (72.2%)	
RESECTABLE	32 (29.6%)	17 (31.5%)	15 (27.8%)	
**HEPATOPATHY PRIOR TO LIVER SURGERY:**				0.188
NONE OR MILD HEPATOPATHY	54 (53.5%)	24 (46.2%)	30 (61.2%)	
MODERATE OR SEVERE HEPATOPATHY	47 (46.5%)	28 (53.8%)	19 (38.8%)	
**LIVER SURGERY PERIOD**				0.162
2005–2013	66 (61.1%)	30 (55.6%)	36 (66.7%)	
2014–2023	42 (28.9%)	24 (44.4%)	18 (33.3%)	
**RESECTED PRIMARY TUMOR AND ALL METASTASES**				0.073
YES (CR0)	95 (88%)	44 (81.5%)	51 (94.4%)	
NO (CR2)	13 (12%)	10 (18.5%)	3 (5.6%)	
**CLINICAL RISK SCORE (FONG et al.** [16]**)—2 CATEGORIES:**				0.461
CLINICAL RISK SCORE 0–2	36 (38.3%)	21 (42.9%)	15 (33.3%)	
CLINICAL RISK SCORE 3–5	58 (61.7%)	28 (57.1%)	30 (66.7%)	

* LNR = lymph node ratio (lymph nodes positive/lymph nodes isolated).

**Table 2 cancers-17-01870-t002:** Preoperative Treatment and Tumor Response Characteristics.

	[ALL] N = 108	Cohort 1 (EGFR INHIBITOR) N = 54	Cohort 2 (VEGF INHIBITOR) N = 54	*p*-Value
**CHEMOTHERAPY SCHEDULE BEFORE LIVER SURGERY:**				0.229
OXALIPLATIN + 5FLUORORURACIL (5FU)	69 (63.9%)	31 (57.4%)	38 (70.4%)	
IRINOTECAN + 5FU	39 (36.1%)	23 (42.6%)	16 (29.6%)	
**NUMBER OF CHEMOTHERAPY CYCLES BEFORE LIVER SURGERY**	7.00 [5.00;11.0]	7.00 [5.00;12.0]	8.00 [6.00;10.0]	0.685
**CHEMOTHERAPY DOSE INTENSITY (%)**	95.0 [90.0;100]	91.5 [87.8;100]	100 [90.0;100]	0.227
**CUMULATIVE DOSE OF OXALIPLATIN (mg/m^2^)**	544 [361;833]	510 [355;784]	604 [427;823]	0.541
**CUMULATIVE DOSE OF IRINOTECAN (mg/m^2^)**	1260 [1076;1791]	1260 [1022;1944]	1368 [1215;1620]	0.606
**NUMBER OF MONOCLONAL ANTIBODY CYCLES BEFORE LIVER SURGERY**	6.00 [4.00;9.00]	6.00 [4.00;10.0]	6.50 [4.00;9.00]	0.823
**MONOCLONAL ANTIBODY DOSE INTENSITY (%)**	100 [100;100]	97.13 [94.76;99.50]	99.91 [99.72;100]	0.013
**EGFR INHIBIDOR RELATED SKIN TOXICITY (CTC-AE):**				
1	24 (44.4%)	24 (44.4%)	0 (.%)	
2	20 (37.0%)	20 (37.0%)	0 (.%)	
3	10 (18.5%)	10 (18.5%)	0 (.%)	
**RECIST RESPONSE CRITERIA:**				0.460
COMPLETE RESPONSE	1 (0.94%)	1 (1.89%)	0 (0.00%)	
PARTIAL RESPONSE	73 (68.9%)	39 (73.6%)	34 (64.2%)	
ESTABLE DISEASE	26 (24.5%)	11 (20.8%)	15 (28.3%)	
PROGRESSION DISEASE	6 (5.66%)	2 (3.77%)	4 (7.55%)	
**CEA BEFORE LIVER SURGERY (ng/mL)**	6.93 [3.04;27.3]	6.47 [2.79;19.5]	8.50 [3.52;30.0]	0.421
**CA 19.9 BEFORE LIVER SURGERY (UI/mL)**	23.2 [10.4;53.0]	16.5 [8.12;26.8]	32.2 [14.2;82.5]	0.003

**Table 3 cancers-17-01870-t003:** Operated Liver Characteristics and Pathological Response.

	[ALL] N = 108	Cohort 1 (EGFR INHIBITOR) N = 54	Cohort 2 (VEGF INHIBITOR) N = 54	*p*-Value
**REMOVED LIVER WEIGHT (grams)**	450 [139;689]	449 [142;662]	451 [147;759]	0.946
**LIVER REMOVED WEIGHT/BODY WEIGHT, PERCENTAGE (%)**	0.62 [0.22;1.01]	0.59 [0.21;0.98]	0.68 [0.24;1.03]	0.681
**PATHOLOGICAL MAJOR SIZE OF LIVER METASTASES (mm)**	29.0 [17.0;53.0]	26.0 [15.0;52.0]	30.0 [20.2;53.8]	0.388
**RESECTION MARGIN DISTANCE (mm)**	0.10 [0.00;1.00]	0.20 [0.00;1.00]	0.00 [0.00;1.00]	0.473
**MEDIAN RESIDUAL TUMOR (%)**	35.0 [15.0;50.2]	37.5 [30.0;54.9]	29.7 [10.0;47.3]	0.068
**MEDIAN RESIDUAL FIBROSIS (%)**	31.5 [14.6;49.1]	40.0 [25.4;53.2]	20.6 [8.07;36.9]	<0.001
**MEDIAN RESIDUAL NECROSIS (%)**	20.0 [7.50;45.2]	16.1 [5.00;25.0]	33.0 [15.0;50.0]	0.001
**PATHOLOGICAL RESPONSE BY RUBBIA–BRANDT CRITERIA:**				0.516
TRG1	3 (2.78%)	2 (3.70%)	1 (1.85%)	
TRG2	36 (33.3%)	17 (31.5%)	19 (35.2%)	
TRG3	22 (20.4%)	14 (25.9%)	8 (14.8%)	
TRG4	46 (42.6%)	21 (38.9%)	25 (46.3%)	
TRG5	1 (0.93%)	0 (0.00%)	1 (1.85%)	
**PATHOLOGICAL RESPONSE BY RUBBIA–BRANDT CRITERIA—2 CATEGORIES:**				0.438
TRG NOT RESPONDERS (TRG4–5)	47 (43.5%)	21 (38.9%)	26 (48.1%)	
TRG RESPONDERS (TRG1–3)	61 (56.5%)	33 (61.1%)	28 (51.9%)	
**PATHOLOGICAL RESPONSE BY POULTSIDES CRITERIA:**				0.003
FIBROSIS < 40%	66 (61.1%)	25 (46.3%)	41 (75.9%)	
FIBROSIS ≥ 40%	42 (38.9%)	29 (53.7%)	13 (24.1%)	
**NECROSIS CATEGORIES:**				0.001
NECROSIS < 40%	77 (71.3%)	47 (87.0%)	30 (55.6%)	
NECROSIS ≥ 40%	31 (28.7%)	7 (13.0%)	24 (44.4%)	

**Table 4 cancers-17-01870-t004:** Univariate and multivariate logistic regression model to predict fibrosis ≥40%.

	Univariate Analysis				Multivariate Analysis			
	N	OR	95% CI	*p*-Value	N	OR	95% CI	*p*-Value
**CA 19.9 AT THE METASTATIC DIAGNOSIS**	88	1.000	0.99, 1.00	0.779				
CA 19.9 BEFORE LIVER SURGERY	81	0.999	0.99, 1.00	0.370				
**CEA AT THE METASTATIC DIAGNOSIS**	107	1.000	1.00, 1.00	0.220				
**CEA BEFORE LIVER SURGERY**	103	0.989	0.97, 1.00	0.089				
**CHEMOTHERAPY SCHEDULE BEFORE LIVER SURGERY:**	108							
IRINOTECAN + 5FU (reference)	39	1						
OXALIPLATIN + 5FU	69	0.869	0.39, 1.93	0.732				
**EGFR INHIBITOR SKIN TOXICITY (CTC-AE)**	54							
1 (reference)	24	1						
2	20	0.846	0.19, 3.70	0.825				
3	10	1.857	0.39, 8.68	0.432				
**CLINICAL RISK SCORE (FONG et al.** [16]**) (CRS)**	94				94			
CLINICAL RISK SCORE 0–2 (reference)	36	1			36	1		
CLINICAL RISK SCORE 3–5	58	0.371	0.15, 0.88	**0.025**	58	0.377	0.14, 0.97	**0.044**
**PORTAL EMBOLIZATION/LIGATION:**	108							
NO (reference)	79	1						
YES	29	0.627	0.25, 1.55	0.312				
**MONOCLONAL ANTIBODY BEFORE LIVER SURGERY:**	108				94			
VEGF INHIBITOR (reference)	54	1			45	1		
EGFR INHIBITOR	54	3.658	1.60, 8.32	**0.002**	49	4.967	1.88, 13.08	**0.001**
**NUMBER OF CHEMOTHERAPY CYCLES BEFORE LIVER SURGERY**	108	1.060	0.98, 1.14	0.125				
**NUMBER OF LIVER METASTASES**	107	0.936	0.85, 1.02	0.169				
**NUMBER OF MONOCLONAL ANTIBODY CYCLES BEFORE LIVER SURGERY**	108	1.033	0.95, 1.11	0.403				
**PRIMARY TUMOR SIDE:**	108				94			
RIGHT COLON (reference)	17	1			14	1		
LEFT COLON	91	3.500	0.94, 13.02	**0.062**	80	4.607	0.85, 24.85	0.076
**RAS_BRAF_MUTATIONS:**	104							
RAS_BRAF_MUTATIONS (reference)	41	1						
NO RAS_BRAF_MUTATIONS	63	3.444	1.41, 8.38	**0.006**				
constant = −2.174								
Formula: log (P/1 − P) = −2.174 + (1.603 × VEGF inh = 0 or EGFR inh = 1) + (−0.974 × CRS 0–2 = 0 or CRS 3–5 = 1) + (1.528 × RIGHT COLON = 0 or LEFT COLON = 1).								
P = 1/1 + *e*(−1 × ((−2.174 + (1.603 × VEGF inh = 0 or EGFR inh = 1) + (−0.974 × CRS 0–2 = 0 or CRS 3–5 = 1) + (1.528 × RIGHT COLON = 0 or LEFT COLON = 1)).								

**Table 5 cancers-17-01870-t005:** Probability Groups for Fibrosis ≥ 40%.

**Right colon**	EGFR inhibitor + CRS 0–2	36.10%
	EGFR inhibitor + CRS 3–5	17.50%
	VEGF inhibitor + CRS 0–2	10.20%
	VEGF inhibitor + CRS 3–5	4.10%
**Left colon**	EGFR inhibitor + CRS 0–2	73.2%
	EGFR inhibitor + CRS 3–5	49.5%
	VEGF inhibitor + CRS 0–2	34.3%
	VEGF inhibitor + CRS 3–5	16.5%
Formula: P = 1/1 + e(−1 × ((−2.174 + (1.603 × VEGF inhibitor = 0 or EGFR inhibitor = 1)		
+ (−0.974 × CRS 0–2 = 0 or CRS 3–5 = 1) + (1.528 × right colon = 0 or left colon = 1)).		

**Table 6 cancers-17-01870-t006:** Multivariant Overall Survival Cox-Regression Model.

MULTIVARIANT MODEL BY POULTSIDES RESPONSE CRITERIA					MULTIVARIANT MODEL BY RUBBIA–BRANDT RESPONSE CRITERIA				
Characteristic	N	HR †	95% CI ‡	*p*-Value	Characteristic	N	HR †	95% CI ‡	*p*-Value
**PATHOLOGICAL RESPONSE BY POULTSIDES CRITERIA**	79			0.014	**RUBBIA_BRANDT_CATEGORIES**	79			0.028
FIBROSIS < 40% (reference)		1			TRG NOT RESPONDERS (TRG4–5) (reference)		1		
FIBROSIS ≥40%		0.488	0.270, 0.880		TRG RESPONDERS (TRG1–3)		0.549	0.321, 0.938	
**HEPATOPATHY PRIOR TO LIVER SURGERY:**	79			<0.001	**HEPATOPATHY PRIOR TO LIVER SURGERY:**	79			<0.001
NONE OR MILD HEPATOPATHY (reference)		1			NONE OR MILD HEPATOPATHY (reference)		1	—	
MODERATE OR SEVERE HEPATOPATHY		2.678	1.535, 4.672		MODERATE OR SEVERE HEPATOPATHY		2.821	1.612, 4.936	
**GENDER**	79			0.019	**GENDER**	79			0.031
MALE (reference)		1			MALE (reference)		1		
FEMALE		0.505	0.281, 0.909		FEMALE		0.531	0.294, 0.960	
**LNR ***	79	4.19	1.222, 14.36	0.033	**LNR ***	79	3.467	1.058, 11.36	0.055
**CEA BEFORE LIVER SURGERY (ng/mL)**	79	1.001	1.000, 1.002	0.059	**CEA BEFORE LIVER SURGERY (ng/mL)**	79	1.002	1.000, 1.003	0.029
**RESECTED PRIMARY TUMOR AND ALL METASTASES**	79				**RESECTED PRIMARY TUMOR AND ALL METASTASES**	79			
YES (CR0) (reference)		1			YES (CR0) (reference)		1		
NO (CR2)		1.065	0.394, 2.876	0.901	NO (CR2)		0.952	0.357, 2.541	0.922
† HR = hazard ratio, ‡ CI = confidence interval * LNR = lymph node ratio (lymph node positive/lymph node isolated)					† HR = hazard ratio, ‡ CI = confidence interval * LNR = lymph node ratio (lymph node positive/lymph node isolated)				
Akaike information index = 407.284					Akaike information index = 408.493				

## Data Availability

The data supporting the findings of this study can be obtained by contacting the corresponding author upon request.

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
