# Peer review of "The Prognostic Value of Tumor Fibrosis in Patients Undergoing Hepatic Metastasectomy for Colorectal Cancer: A Retrospective Pooled Analysis"

_cancers, 2025, doi:10.3390/cancers17111870_

Round 1
Reviewer 1 Report
Comments and Suggestions for Authors
Please refer to my comments:
- Please adhere to the journal typesetting and formatting guidelines in preparing the manuscripts and tables. Be consistent with font size and type, spacing & etc. Same goes with referencing style and format writing. Besides, you should have an authorship contribution and an ethical statement after the conclusion. Please refer to the template again
- Please include ethical approval date and approval number.
- Description for statistical analysis is missing in the Materials and Methods
- The 18-year inclusion period (2005–2023) may introduces potential bias due to evolving clinical practices in these 20 years. Please justify why need to cover this huge period? Please include the range in year of hepatic surgery (like 2010-2023) in Table 1.
- There is significant imbalance in RAS/BRAF mutation status between the cohorts, where cohort 1 had 92.6% wild-type vs. 26% in Cohort 2. This may introduce confounding in analysis. What is the author’s justification?
- We can observe an increase in fibrosis with anti-EGFR therapy, but this does not translate into a survival benefit. The statement that anti-EGFR therapy is associated with greater fibrosis should not be interpreted as a clinical recommendation without a stronger survival correlation. How reliable is fibrosis outcome as a surrogate endpoint here?
- I am expecting the data that you discussed, like extrahepatic metastasis, MMR deficiency or BRAF mutation, shall be included in Table 1 or at least in supplementary tables
Author Response
I highlighted the suggested changes in red. Thank you for your comments.

Reviewer 2 Report
Comments and Suggestions for Authors
The manuscript titled “The Prognostic Value of Tumor Fibrosis in patients undergoing Hepatic Metastasectomy for colorectal cancer: A Retrospective Pooled Analysis” by Xavier Hernández-Yagüe et al, presents a comparative analysis of neoadjuvant chemotherapy (NAC) combined with anti-EGFR versus anti-VEGF therapy, exploring their respective associations with fibrosis in liver metastases from metastatic colorectal cancer (mCRC) and related clinical outcomes. The study further examines the prognostic utility of the Poultsides classification in relation to the two-category Rubbia-Brandt classification. The study demonstrates methodological rigor, including an adequately powered study design, clear descriptions of the sample types, and the comparability of baseline characteristics between the NAC + anti-EGFR and NAC + anti-VEGF groups. These strengths enhance the credibility of the findings.
Below are a few minor suggestions:
- In the abstract, under the conclusion, the statement “Anti-EGFR therapy is associated with greater fibrosis than anti-VEGF” appears to generalize the findings. As the multivariate analysis was limited to RAS wild-type cases, and the univariate model alone suggested this association across the entire cohort, this conclusion may risk overinterpretation. It would be more accurate and appropriately cautious to indicate that this association was observed primarily within the RAS wild-type subgroup based on multivariate modeling.
- Consider providing a supporting reference to substantiate the sentence “Notably, multivariate analyses suggest that TRG is an independent prognostic factor for both Disease-Free Survival (DFS) and Overall Survival (OS)” on page 2, 3rd paragraph under ‘Introduction’.
- On page 15, section 3.2.3, although the model was developed using existing data, either internal or external validation is warranted to assess the predictive performance of the formula. We suggest including a statement acknowledging this need for validation and, proposing it as a next step for future work.
Author Response
I highlighted the suggested changes in blue. Thank you for your comments.

Reviewer 3 Report
Comments and Suggestions for Authors
The main aims of the retrospective study on metastatic colorectal cancer include: verifying whether chemotherapy with anti-EGFR induces a higher degree of fibrosis than chemotherapy with bevacizumab, and whether fibrosis >=40% translates into better survival. The analysed cohort consisted of patients who underwent liver resection between 2005 and 2023.
The prognostic value of fibrosis in liver metastases has not been unequivocally established so far. Therefore, this study is interesting from the clinical point of view.
There are some issues that should be clarified or corrected before possible publication of this manuscript:
1) One of the inclusion criteria is radical surgery for hepatic metastases (R0), regardless of whether surgery for extrahepatic metastases was performed. When extrahepatic metastases were not resected, it was not R0 surgery, and the patients were treated without curative intent. Extrahepatic metastases were diagnosed in 25.9%. Inclusion of these patients leads to selection bias, and survival associated with fibrosis in liver metastases is influenced by the presence of extrahepatic metastases. My suggestion is to perform analyses only in patients without extrahepatic metastases. At least R0, R1 and R2 resections should be reported for each analyzed group.
2) In the paragraph entitled "Statistic" the authors stated that for survival analysis, Kaplan-Meier curves would be generated and compared using the log-rank test. There are no Kaplan-Meier curves in the manuscript that I received for review. As survival is one of primary outcomes, survival curves should be included in the paper.
3) Why siginificatnly higher fibrosis in cohort 1 did not translated into survival benefit? In Cox regression model fibrosis improved survival. This finding was emphasized in conclusion but was not elaborated in the discussion section.
4) In tables, the grey background is unnecessary as they are clearly readable and the headings are bolded.
5) In Table 1 the ALL column seems to be unnecessary, the same the last column (N). In the same table there is serious discrepancy - pT and pN values are reported and below ypT and ypN. As all patients were operated on after chemotherapy only yp staging should be reported. Moreover, ypT and ypN categories are reported only in 43 out of 108 patients. Does it mean that in 65 patients the primary tumor was not resected? It should be clarified in the table and in the manuscript.
6) In chapter 3.1.1, the authors stated that the primary tumor was resected in 94.4% of patients. Patients without primary tumor resection should be excluded from analysis due to worse prognosis and possible bias. It contradicts the inclusion criterion that R0 resections were analysed.
7) Table 2 - column ALL and N may be deleted, grey background should be removed. Why RECIST criteria were reported for 106 pts not 108?
8) Chapter 3.1.4 - The authors stated that significant differences had been observed in the administration of adjuvant therapy following hepatic surgery or first-line therapy for patients with unresected metastases. Again, it should be specified whether the analysis includes only R0 resections. After R0 resection, recurrence-free survival or disease-free survival should be reported, not progression-free survival PFS. PFS suggests that the neoplasm was not radically resected.
9) Table 4 - if the table should be included in the manuscript, the lower and upper bounds of CI should be reported with two decimal places. The column coefficient for multivariable regression should be removed.
10) In chapter 3.3.2, there is a reference to Table 6, but such a table is not included in the manuscript.
11) In chapter 3.1.4 - The authors stated that significant differences had been observed in the administration of chemotherapy following hepatic surgery. As chemotherapy may influence survival, the Cox models should be adjusted for this parameter. Please include postoperative chemotherapy in the Cox regression.
12) Was proportional hazard assumption verified? Which test was used for this purpose?
13) In Table 5 and other tables, the abbreviation LNR should be explained (lymph node ratio) below the table.
14) In discussion, there is a sentence: 'Additionally, the presence of a left-sided pri-mary tumor was also associated with a higher likelihood of fibrosis ≥40%, though this association was less statistically significant, as evidenced by an OR of 4.607 (95% CI: 0.854–24.855; p=NS). '. The association was not statistically significant! It cannot be written as 'not less significant'.
Author Response
I highlighted the suggested changes in red. I prepared the Kaplan–Meier tables and incorporated the suggested changes.Thank you for your comments.

Round 2
Reviewer 3 Report
Comments and Suggestions for Authors
The Authors clarified a lot of issues in the revised version of the manuscript and improved the quality of the tables. Although they explained the differences between pTNM and ypTNM in Table 1, it is still unclear in this table. There is a rule that tables in scientific manuscripts should be self-contained. Please correct Table 1 so that it is unequivocally stated what pTNM and ypTNM refer to. I strongly suggest omitting the last column in the table as the numbers of patients are specified in other columns and headings.
After these corrections, the manuscript may be considered by the Editors for publication.
Author Response
Comment Reviewer 3: The Authors clarified a lot of issues in the revised version of the manuscript and improved the quality of the tables. Although they explained the differences between pTNM and ypTNM in Table 1, it is still unclear in this table. There is a rule that tables in scientific manuscripts should be self-contained. Please correct Table 1 so that it is unequivocally stated what pTNM and ypTNM refer to. I strongly suggest omitting the last column in the table as the numbers of patients are specified in other columns and headings
Response:
First of all, I would like to thank you for the comments and suggestions provided during the first and second rounds of manuscript review. I believe that all the comments have undoubtedly helped improve the final manuscript.
Following the suggestion from the last review, I have clarified the pTNM and ypTNM status in Table 1. Additionally, I have removed the last column from the descriptive tables (included table 1) where the data was redundant with the previous columns. I am now resubmitting all the corrected tables.
Best regards.
